

# Diurnal variation in middle atmospheric ozone by ground based microwave radiometry at Ny-Ålesund over 1 year

Franziska Schranz[1], Susana Fernandez[2], Niklaus Kämpfer[1], and Mathias Palm[3]

[1]Institute of Applied Physics, University of Bern, Bern, Switzerland
[2]Department of Physics, University of Santiago de Chile, Santiago, Chile
[3]Institute of Environmental Physics, University of Bremen, Bremen, Germany

*Correspondence to:* Franziska Schranz (franziska.schranz@iap.unibe.ch)

**Abstract.** We present an analysis of the diurnal ozone cycle from one year of continuous ozone measurements from two ground based microwave radiometers in the Arctic. The instruments GROMOS-C and OZORAM have been located at the AWIPEV research base at Ny-Ålesund, Svalbard (79° N, 12° E) and gathered a comprehensive time series of middle atmospheric ozone profiles with a high time resolution. An intercomparison was performed with EOS MLS and ozone sonde measurements and simulations with SD-WACCM. The measured datasets were used to study the photochemically induced diurnal cycle of ozone in the stratosphere and mesosphere. Throughout the year the insolation in the Arctic changes drastically from polar night to polar day. Accordingly, the seasonal variations in the diurnal ozone cycle are large. In the stratosphere we found a diurnal cycle throughout the whole period of polar day with the largest amplitude in April. In the mesosphere a diurnal cycle was detected in spring and fall. SD-WACCM has proved to well capture the diurnal cycle and was therefore used to analyse the chemical reaction rates of ozone production and loss at equinox and summer solstice. Furthermore GROMOS-C proofed capable of measuring the tertiary ozone layer above Ny-Ålesund in winter.

## 1 Introduction

Ozone is a molecule with important tasks in the earth atmosphere. It protects the biosphere from harmful UV radiation and controls the thermal structure of the stratosphere. In the middle atmosphere the ozone abundance is governed by transport processes and (photo-)chemical reactions and shows variations on diurnal to interannual time scales. The diurnal variations of ozone are mainly controlled by photochemical and catalytic processes. The basic pattern of the diurnal variation from the stratosphere to the mesosphere is well described by a set of pure oxygen reactions derived by Chapman (1930).

$$O_2 + h\nu \quad \rightarrow \quad O + O \tag{R1}$$

$$O + O_2 + M \quad \rightarrow \quad O_3 + M \tag{R2}$$

$$O_3 + h\nu \quad \rightarrow \quad O_2 + O \tag{R3}$$

$$O_3 + O \quad \rightarrow \quad O_2 + O_2 \tag{R4}$$

During daytime, when the atmosphere is sunlit, odd oxygen ($O_x$ = O + $O_3$) is produced by reaction R1 and is partitioned between atomic oxygen and ozone by the reactions R2 and R3. The ozone production (R2) depends on the abundance of a third



body M and therefore on pressure. In the stratosphere this reaction is very efficient relative to R3 and all odd oxygen is in the form of ozone which forms a maximum in the late afternoon. With increasing altitude reaction R3 gets less efficient relative to R2 and more odd oxygen is stored as atomic oxygen during daytime. After sunset all the atomic oxygen recombines which leads to a night time maximum of ozone. The night to day ratio of ozone in the mesosphere is increasing with altitude. For

a realistic description of the ozone cycle additional destruction processes and transport effects have to be taken into account. This includes catalytic ozone depletion by odd hydrogen ($HO_x$), odd nitrogen ($NO_x$) and inorganic halogens (Cl, Br). The magnitude of the diurnal cycle in the different altitude regimes is determined by the solar zenith angle and the length of the day. At high latitudes the diurnal cycle is highly depending on the season because the insolation conditions vary strongly throughout the year. The period of polar night, the period of polar day and the intermediate periods with light and darkness in

one day characterise the pattern of the diurnal cycle in the Arctic.

The diurnal cycle of ozone in the mesosphere has been vastly studied by means of ground based microwave instruments (Penfield et al., 1976; Wilson and Schwartz, 1981; Lobsiger and Künzi, 1986; Zommerfelds et al., 1989; Connor et al., 1994; Palm et al., 2013) and other techniques (Vaughan, 1982; Ricaud et al., 1996). A sharp decrease in ozone volume mixing ratio (VMR) after sunrise of up to 65% at 0.1 hPa was reported. After sunset the night time values are restored quickly.

Observations from ground based ozone radiometers from the Network for the Detection of Atmospheric Composition Change (NDACC) were used to study the diurnal cycle in the stratosphere. Haefele et al. (2008) found an afternoon enhancement in the observations of the ground based microwave radiometer SOMORA at the Alpine station in Payerne, Switzerland. This enhancement is also seen in the summertime observations from OZORAM at the Arctic station at Ny-Ålesund, Svalbard (Palm et al., 2013). A monthly climatology of the diurnal ozone cycle was derived from 17 years of observations of GROMOS

which is operated at Bern, Switzerland (Studer et al., 2014). Parrish et al. (2014) used 19 years of observations by a ground based microwave radiometer at the Mauna Loa Observatory, Hawaii to analyse the diurnal cycle. Both studies confirm the afternoon enhancement of ozone in the stratosphere.

Satellite born observations were also used to study the diurnal cycle of ozone. Huang et al. (2010) analysed measurements of UARS MLS , which cycles through 24 hours over a given location within 36 days, and derived the diurnal cycle for latitudes

between 40°N and 40°S. Measurements from the Superconducting Submillimeter-Wave Limb-Emission Sounder (SMILES) carried by the International Space Station, which samples 24 hours in 60 days, were used by Sakazaki et al. (2013) to analyse the diurnal variations in the same latitude band. Both studies show that the relative amplitude of the afternoon maximum with respect to the minimum value at 3 hPa during summer are around 3% in the tropics and around 7% in the midlatitudes. The global behaviour of the stratospheric ozone cycle at 5 hPa and a detailed analysis of the contributing chemical reactions are

discussed in a model study by Schanz et al. (2014a).

Measurements of the diurnal cycle in the Arctic are rare. Palm et al. (2013) published a time series of stratospheric ozone measured by OZORAM which covers three intervals of several days during summer 2010 and reported the existence of a diurnal variation during polar day. The monthly mean diurnal cycle measured by OZORAM in June 2011 was intercompared to model reanalyses and revealed a relative amplitude of 8% at 5 hPa (Schanz et al., 2014b).



We present an analysis of the diurnal cycle in the arctic stratosphere and mesosphere for different insolation conditions throughout the year. The ozone time series used for this study were measured by the two ground based microwave radiometers GROMOS-C and OZORAM which are located at Ny-Ålesund, Svalbard (79° N, 12° E) in the Arctic. It is a comprehensive dataset of arctic-, middle atmospheric ozone with a high time resolution. The chemical and photochemical reactions that lead to a diurnal cycle during polar day are studied with the atmospheric model SD-WACCM. The datasets are further intercompared to measurements with balloon born ozone sondes and MLS and to the simulations with SD-WACCM.

The remainder of this paper is organized as follows: Section 2 introduces the measurement campaign at Ny-Ålesund. The two ground based microwave radiometers as well as the other instruments and the model used for intercomparison are described in section 3. Section 4 compares the microwave radiometer measurements to measurements by MLS and balloon born ozone sondes and to SD-WACCM simulations. The diurnal cycle of ozone is discussed in section 5 and measurements of the tertiary ozone maximum are presented in section 6. Section 7 summarizes the content and draws the conclusions.

## 2 Ny-Ålesund Campaign

A measurement campaign with ground based microwave radiometers for ozone and water vapour has been taking place at the AWIPEV research base at Ny-Ålesund, Svalbard (79° N, 12° E) since September 2015 and is ongoing. The campaign is a cooperation of the University of Bremen and the University of Bern. The University of Bremen contributes with the NDACC instrument OZORAM (Ozone Radiometer for Atmospheric Measurements) and the University of Bern contributes with the two instruments MIAWARA-C (Middle Atmospheric Water Vapour Radiometer for Campaigns) and GROMOS-C (Ground-based Ozone Monitoring System for Campaigns). Due to its high latitude Ny-Ålesund is the ideal location for observations inside the polar vortex and to study events related to its dynamics, like sudden stratospheric warmings. The scientific focus of the campaign lies on the investigation of dynamical events, the link between middle atmospheric ozone and water vapour chemistry and the analysis of the temporal variability of those atmospheric constituents. The present study covers the temporal variability of ozone.

Svalbard is an archipelago in the Arctic ocean and the research settlement Ny-Ålesund is located in the North-West of the main island Spitsbergen at the shore of the Kongsfjorden bay. It is one of the northernmost permanent settlements and has been a base for arctic expeditions and a host to research stations since the early 1900. At the moment it hosts 15 permanent research stations from 10 different countries mainly focusing on earth and environmental sciences (www.kingsbay.no). Among them is the AWIPEV base, a joint French-German research station operated all year round (www.awipev.eu). Polar day and Polar night last for 4 month each. This allows to study the ozone (photo-)chemistry during day and night conditions. Despite the long winter the climate at Ny-Ålesund is mild because of the influence of the North Atlantic Current. The result is a high humidity of the atmosphere compared to other Arctic locations. This affects remote sensing observations of the middle atmosphere where the transparency of the troposphere has to be taken into account. For microwave radiometry as described in this paper the average optical depth at 110.8 GHz was 0.7 in winter and almost doubled to 1.2 in summer. An opacity value lower than 0.5 is considered to be very good for such observations (Fernandez et al., 2016).





During the first winter of the campaign a strong and stable polar vortex system formed and stratospheric temperatures were very low. Accordingly the area of polar stratospheric clouds was large and a considerable amount of active chlorine (above 3 ppb) was present in January (WMO Arctic Ozone Bulletin). The subsequent ozone loss in spring was however terminated in early March by a sudden stratospheric final warming (Manney and Lawrence, 2016).

## 3 Instruments and Model

### 3.1 GROMOS-C

The microwave radiometer GROMOS-C (GROund based Ozone MOnitoring System for Campaigns) was built at the University of Bern. An impression of the instrument, which is currently located at the AWIPEV research base at Ny-Ålesund, is given in Figure 1. The design of GROMOS-C is very compact, it only needs power and internet connection and it is remotely controlled making it a valuable instrument for campaigns. Detailed information about the instrument can be found in Fernandez et al. (2015).

The concept of ground based microwave radiometry is to spectrally resolve the pressure broadened emissions lines of atmospheric constituents and to retrieve an altitude distribution of these constituents with an optimal estimation method. GROMOS-C is able to measure at different frequencies: The 110.8 GHz ozone emission line is measured by default but it can also switch to the 115.3 GHz emission line of CO. Additionally it is able to retrieve wind profiles with the method developed by Rüfenacht et al. (2012). To retrieve the ozone profiles from the calibrated GROMOS-C spectra the Atmospheric Radiative Transfer Simulator version 2 (ARTS2) (Eriksson et al., 2011) is used together with Qpack2 (Eriksson et al., 2005). The retrieval algorithm implemented in Qpack2 is based on the optimal estimation method of Rodgers (1976). The settings for the GROMOS-C retrieval are described in Fernandez et al. (2015). The results of a validation campaign at the NDACC station of La Réunion island (Fernandez et al., 2016) show a good performance of GROMOS-C. In the altitude range of 25-60 km the relative difference to MLS observations lies within 5%.

From the GROMOS-C spectra two-hourly profiles with an altitude range of 23–70 km are retrieved in the basic mode. The vertical resolution of the profiles, which is defined by the width of the averaging kernels, is 10–12 km in the stratosphere and increases up to 20 km in the mesosphere. The measurements are performed under an elevation angle of 22° in the four cardinal directions (N-E-S-W) which allows to have observations in and outside of the polar vortex if the edge of the vortex is close to Ny-Ålesund. This measurement concept of GROMOS-C is unique in microwave radiometry. GROMOS-C had the chance to measure different ozone concentrations in and outside of the polar vortex during a vortex split in the beginning of November 2016. The analysis data from ECMWF (European Center for Medium Range Weather Forecast) show that Ny-Ålesund was lying inside one leg of the vortex and the edge of the vortex was approaching from West/South-West (Fig. 2, bottom). GROMOS-C measured enhanced ozone VMR outside of the vortex first in westward direction and then also in southward direction while the ozone VMR stayed constant inside the vortex in north and eastward direction (Fig. 2, top).



### 3.2 OZORAM

OZORAM (OZOne Radiometer for Atmospheric Measurements) is a ground based microwave radiometer built at the University of Bremen. It is located at the AWIPEV research base at Ny-Ålesund since 1994 and is in its current observation mode since 2008. OZORAM is an instrument of the Network for the Detection of Atmospheric Composition Change (NDACC). It measures the ozone emission line at 142.2 GHz and provides profiles in the altitude range of 25–70 km with a time resolution of 1 hour and a vertical resolution of 10–20 km. The observations are performed under an angle of 20° elevation and 113° azimuth. For the retrievals of the spectra recorded by OZORAM ARTS 1.1 and QPACK 1 is used. For a detailed description of the instrument and the measurements refer to Palm et al. (2010). The measurements of OZORAM are available at the NDACC data repository ftp://ftp.cpc.ncep.noaa.gov/ndacc/station/nyalsund/hdf/mwave/.

### 3.3 MLS

EOS MLS is the Earth Observing System Microwave Limb Sounder on board of NASAs Aura satellite. The Aura satellite is in a sun synchronous orbit at 705 km altitude with 98° inclination and a period of 98.8 minutes. It overpasses a location at the Earth surface two times a day at fixed times. MLS scans the limb in direction of orbital motion which gives an almost pole to pole coverage (82°S–82°N) for every orbit. The separations of two adjacent limb scans is 165 km along-track (1.5° on the great circle). Detailed information about EOS MLS is provided in Waters et al. (2006).

MLS measures ozone at 240 GHz. The version of the retrieval algorithm used for this study is v4.2. MLS provides ozone profiles from 12–80 km. From the upper troposphere to the mid mesosphere the vertical resolution is 2.7–3 km. In the upper mesosphere the vertical resolution increases up to 5 km (Froidevaux et al., 2008). Due to the sun synchronous orbit of the Aura satellite the measurements above Ny-Ålesund are performed twice a day within intervals of 1.5 hours around 5 am and 11 am CET. For the intercomparison with the instruments located at Ny-Ålesund MLS data are taken if the location of the measurement is within $\pm 1.2°$ longitude and $\pm 6°$ latitude from Ny-Ålesund. This corresponds to a square area with a side length of 260 km centred at Ny-Ålesund.

### 3.4 Ozone Radiosonde

At the AWIPEV research base balloon-born ozone sondes are launched regularly once per week. During the polar night, the frequency is increased to at least two sondes per week. Ozone is measured with an electrochemical concentration cell (ECC) model 6A. A pump efficiency correction is applied to the measured ozone profile. The altitude reached by the sonde is about 30 km and measurements are performed every 5 seconds during the ascent of the balloon which takes about 1 hour and 40 minutes. This leads to an average altitude resolution of 30 m. The radiosonde data are available at the NDACC data repository ftp://ftp.cpc.ncep.noaa.gov/ndacc/station/nyalsund/ames/o3sonde/.



## 3.5 SD-WACCM

The Whole Atmosphere Community Climate Model (WACCM) is a coupled chemistry climate model and a component set of the Community Earth System Model (CESM) version 1.2.2. It was developed at the National Center for Atmospheric Research (NCAR) and is based on the Community Atmosphere Model (CAM) (Collins et al., 2006). The chemistry module

is taken from the Model for Ozone and Related Tracers (MOZART) (Emmons et al., 2010). WACCM has the capability to run in specified-dynamics mode called SD-WACCM (Lamarque et al., 2012; Kunz et al., 2011; Brakebusch et al., 2013). The meteorological fields of the model are thereby constrained by measurements to ensure a most realistic modelling of the dynamics and temperature. This allows for a comparison of the model output with measurements of the atmospheric constituents for a specific time period. Without the nudging a comparison would only be possible in a statistical sense. The

utility of SD-WACCM simulations have been shown by Hoffmann et al. (2012) who compared CO simulated by SD-WACCM and ground based millimeterwave measurements of CO above Kiruna, Sweden.

The altitude range of SD-WACCM is 0–145 km. The grid has 88 levels with an altitude resolution of 0.5–4 km. The horizontal resolution is 1.9° latitude x 2.5° longitude and the internal time step is 30 min. The fields that are constrained in the specified dynamics mode are temperature, horizontal wind, surface wind stress, surface pressure, specific and latent heat fluxes. At ev-

ery time step the fields are nudged by the meteorological analysis fields of the Goddard Earth Observing System 5 (GEOS5) (Rienecker et al., 2008). The strength of the nudging is 10% up to 50 km altitude and then linearly decreasing to 0 % at 60 km.

## 4   Intercomparison

In this section the measurements from the ground based ozone microwave radiometers GROMOS-C and OZORAM are inter-compared to the measurements of MLS and radio sondes and to simulations with SD-WACCM.

Figure 3 shows the ozone time series measured by GROMOS-C during the first year of the Ny-Ålesund campaign. Data gaps are indicated by white vertical lines. During winter they correspond to periods when GROMOS-C switched to carbon monoxide measurements. In summer the data gaps are due to very high opacity values and strong precipitation which caused the retrieval process to fail. During the winter half year the ozone layer is dominated by the dynamics of the polar vortex. Ozone volume mixing ratios decrease sharply when the vortex passes over Ny-Ålesund at a given altitude. The vortex shift of

early February and the final stratospheric warming of March (Manney and Lawrence, 2016) led to increases in stratospheric ozone with maximal values of 7 ppm. In summer stratospheric ozone is mostly dominated by photochemistry. Figure 4 shows the simulation of ozone by SD-WACCM at Ny-Ålesund for the same period. The white lines indicate the polar day and night terminators respectively. At an altitude of 70 km the polar night lasts 2.5 month where at 10 km it lasts 3.5 month. The period with daily sunrise and sunset is with 2 month almost the same at all altitudes. In winter Fig. 4 clearly shows all three ozone

layers. The main ozone layer at an altitude of 35 km, the secondary ozone layer at 95 km and the tertiary ozone layer at 70 km. SD-WACCM captures the dynamical features of ozone in the main layer very well. The data also show that the secondary ozone layer persists during polar day even though very faint. Remarkable are the two sudden increases in ozone in the second ozone maximum during the sudden stratospheric final warming in the beginning of March. These two increases seem to be



connected to an ozone decrease in the tertiary ozone maximum and an increase in stratospheric ozone. However the reason of this phenomena is not known to us and would merit its own investigation. The photochemically induced diurnal variations start shortly after the ending of the polar night at all altitudes. In the stratosphere diurnal variations are seen throughout the polar day whereas in the mesosphere there is no diurnal variation during polar day.

In order to compare the retrieved profiles with the high resolution profiles of MLS, SD-WACCM and the ozone sondes, they are convolved with the averaging kernels of the microwave radiometers. The convolution is performed according to

$$x_{conv} = x_a - \mathbf{A}(x_a - x) \tag{1}$$

where $x_{conv}$ is the convolved profile, $x_a$ is the apriori profile, $\mathbf{A}$ is the averaging kernel matrix and $x$ is the high resolution profile. For the apriori profile ozone data from a MLS climatology for the years 2004–2013 with monthly mean values are

taken. Daily mean values of the convolved time series are taken for the intercomparison. The dataset is then averaged over 3 pressure intervals. The intervals cover the region from 50 hPa to 0.1 hPa and correspond to the middle stratosphere, the upper stratosphere and the lower mesosphere. Ozone sonde measurements are available for the lowest pressure interval only and all individual measurements are displayed. Figures 5 and 6 show the ozone time series and the relative differences of the microwave radiometers and the convolved datasets. The periods of polar day and night and the intermediate period are

indicated by coloured backgrounds.

In the mesosphere (0.1-1 hPa) GROMOS-C is mostly within 20 % of the MLS and SD-WACCM data and has an offset of 10 %. In the middle and upper stratosphere (10-50 hPa and 1-10 hPa) the relative difference of GROMOS-C and MLS is mostly within 10 % during winter. An exception is October 2015 where GROMOS-C measured too low ozone VMR in the middle stratosphere. In summer GROMOS-C starts to overestimate ozone with 10 %. Compared to the ozone sonde GROMOS-C is

overestimating ozone during the whole year by 10 %.

In the mesosphere OZORAM has an agreement with MLS and SD-WACCM within 10 % except during winter where it is underestimating ozone by 20 %. In the upper stratosphere all data sets are within 10 %. However in the middle stratosphere OZORAM is underestimating ozone in winter by up to 20 %. During summer it is within 10 % of the MLS measurements again.

Figure 7 shows averaged ozone profiles and the relative differences of GROMOS-C and OZORAM to the convolved MLS and SD-WACCM profiles. The profiles were averaged during a period in winter and in summer. This comparison shows that the retrieval of OZORAM oscillates during winter but has a good agreement to MLS (within 5 % from 100-0.2 hPa) during summer. For GROMOS-C it is the other way around. During winter the average agreement to MLS is within 5 % from 70-0.5 hPa. However during summer an offset to MLS is detected but it is still within 10 %.

The difference in the performance of the two radiometers from winter to summer measurements might be caused by the different treatment of the troposphere. A tropospheric correction has been applied to the GROMOS-C spectra, such that the retrieval starts at the tropopause level according to Fernandez et al. (2015). This treatment of the troposphere proved to be robust (Fernandez et al., 2016). For OZORAM an other retrieval concept is used where the opacity of the troposphere is retrieved




together with the ozone profile. In order to do this, standard $H_2O$ and $O_2$ profiles are scaled to match the radiation background in the measured region.

## 5 Diurnal Cycle of Ozone

In this section we analyse the diurnal cycle of middle atmospheric ozone above Ny-Ålesund. We use one year of measurements

from the ground based microwave radiometers GROMOS-C and OZORAM, as well as ozone simulations from SD-WACCM The characteristic patterns of the diurnal cycle are shown for the different insolation conditions throughout the year. Furthermore the SD-WACCM model is used to separate the contributions from different reaction paths to the daily ozone production and loss.

An overview of the diurnal cycle above Ny-Ålesund is given in Fig. 8 (top). It shows SD-WACCM ozone VMR over the

course of one year and at three pressure levels. During winter ozone encounters strong fluctuations due to the dynamics of the polar vortex. In the stratosphere these fluctuations are strongest in fall and spring during the formation and the break down of the polar vortex. Ozone changes of up to several ppm per day are seen in the spring and fall period. This was already observed in a model study by Schanz et al. (2014a, Fig. 6). In spring 2016 exceptionally large changes in ozone VMR were detected which were caused by the sudden stratospheric final warming. During summer the ozone fluctuations decline and the diurnal

cycle is clearly visible. In the mesosphere (at 0.1 hPa) the diurnal cycle is strongest during spring and fall. It goes along with the transition from winter time ozone levels ($\sim$1.7 ppm) to the very low summer time ozone levels, and vice versa. The amplitude of the diurnal cycle is around 1 ppm and the depletion and reformation of ozone are very rapid processes . In the stratopause region and in the stratosphere the diurnal cycle goes on during the whole period of polar day. In the stratopause region (at 1 hPa) the amplitude of the diurnal cycle is largest in the beginning of May with 0.55 ppm. The stratosphere (at 10 hPa) shows

the largest amplitudes around summer solstice with 0.25 ppm (Fig. 8, bottom). During polar night there is no diurnal cycle present at all three pressure levels. From winter to summer the geopotential height from SD-WACCM for a constant pressure level varies about 7.5 km in the mesosphere and 5.5 km in the stratosphere. This is smaller than the altitude resolution of the microwave instruments.

For a detailed analysis we look at the averaged diurnal cycle for five time intervals. Because the insolation conditions are

symmetric around summer solstice these intervals are taken between February and July 2016 (Fig. 9-13). The time intervals have different lengths depending on the time of the year. From the end of the polar night the length of day time increases rapidly (23 minutes per day on average) until the polar day starts. To get a reasonable signal to noise ratio of the diurnal variation in the measurements we choose intervals of 10 days. The shift of the sunrise and sunset time over these 10 days is about 2 hours. This implies that the flanks of the depletion at noon are smudged which is however not critical for our basic statement which

is the characterisation of the seasonal pattern of the diurnal cycle. During the polar day the changes in the daily minimum solar zenith angle are small and we choose intervals of up to 42 days. For all these intervals we calculate the averaged relative difference to the mean night time ozone VMR according to

$$\Delta O_3 = \frac{O_3 - O_{3,night}}{O_{3,night}}, \tag{2}$$




where $O_{3,night}$ is the mean over all measurements performed between 10 pm and 2 am local solar time in one time interval. Figures 9–13 show the relative difference to the mean night time ozone VMR for GROMOS-C (1st panel), OZORAM (2nd panel) and the unconvolved SD-WACCM profiles (3rd panel). The diurnal cycle for four pressure levels is additionally shown for the convolved SD-WACCM data (right). The horizontal black lines indicate the altitude of the corresponding pressure levels

and the black line in the 3rd panel indicates the average sunrise and sunset time. The error bars are the standard error of the mean. GROMOS-C is capable of measuring in the four cardinal directions with an observation angle of 22° elevation. If not stated otherwise, the eastward direction of the GROMOS-C measurements is used for the analysis. OZORAM is observing under an elevation angle of 20° in south-eastern direction (113° azimuth). The SD-WACCM ozone data are taken from the grid point 78.6°N/ 12.5°E which is closest to Ny-Ålesund. To compare the SD-WACCM simulation with the ground based

ozone measurements the data are convolved with the averaging kernels of GROMOS-C. Since the vertical resolution of the SD-WACCM ozone profiles is degraded by the convolution also the unconvolved SD-WACCM profiles are shown.

In the *mesosphere* the instruments and the model observe the sharp transitions from night time ozone VMR to day time VMR during the period between polar night and day. This transition is caused by the repartitioning of odd oxygen after sunrise. The dip in ozone VMR is deep and narrow in mid February (70 %) when the sun is above horizon for about 8 hours. One month

later the sun stays above horizon for 16 hours and the period with depleted ozone is wider accordingly. In mid April the polar day has already started in the mesosphere. Remarkable is that the recovery of ozone at midnight is still present. During the summer months there are no diurnal ozone variations detected in the mesosphere. Around equinox the relative difference of the day to night time ozone abundance is about 65 %. Studies from the midlatitudes using measurements by UARS MLS (Ricaud et al., 1996) and a ground based microwave radiometer (eg. Parrish et al., 2014) find the same relative decrease of ozone.

The *stratopause* region lies in between the mesosphere and the stratosphere at about 1 hPa and the diurnal cycle is influenced by both regimes. In February a decrease in ozone VMR around noon is detected. In March the diurnal cycle shows the characteristic behaviour of the stratosphere: an ozone minimum in the morning and a maximum in the afternoon. In April the pressure level of 1 hPa lies in between the mesospheric regime with a depletion of ozone during daytime and the stratospheric regime with a morning minimum and an afternoon maximum. For SD-WACCM this results in two ozone maxima at 8 am and at 9 pm.

The instruments can not fully resolve the first maximum since it is smaller in vertical extent than the width of the averaging kernel. In May and June/July the depletion of ozone during day dominates. An ozone maximum around midnight is followed by a minimum around noon. The relative amplitude of the diurnal cycle is largest in May (8 %) and decreases towards summer solstice (5 %). At that pressure level SD-WACCM shows generally larger amplitudes of the diurnal cycle than the microwave instruments. The convolution of the model data with the averaging kernels of GROMOS-C smooths the profiles and brings the

simulation closer to the measurements.

In the *stratosphere* a robust diurnal cycle is identified during the whole period of polar day. At 3 hPa a diurnal variation is already seen in February and at 10 hPa it starts in March. During these two months ozone has a maximum around sunrise and is decreasing towards sunset. In March the errorbars at 10 hPa are large with ±10 %. This might be caused by the sudden stratospheric warming which lead to an enhanced ozone variability. From April on the distinct morning minimum and afternoon

maximum is seen. At 10 hPa the relative amplitude is largest at summer solstice (7 %) where at 3 hPa it is largest in April (13



%). The existence of a diurnal ozone cycle during polar day indicates that the variation of the solar zenith angle over a day is enough to create a change in the ratio of production and loss rates. Schanz et al. (2014b) used OZORAM and WACCM data to analyse the diurnal cycle above Ny-Ålesund in June 2011. The study found a relative amplitude of the diurnal ozone cycle of 8 % at 5 hPa. For summer solstice and 5 hPa GROMOS-C measures a relative amplitude of the diurnal cycle of 7 %. Previous studies with ground based microwave radiometers in the mid latitudes also show a diurnal cycle in the stratosphere. However, the studies from Bern and Payerne, Switzerland (47°N) (Studer et al., 2014; Haefele et al., 2008) show the typical morning minimum and afternoon maximum only below 10 hPa. At 5 hPa Studer et al. (2014) got a relative amplitude of 7 %. At Mauna Loa, Hawaii (19.5°N) the characteristic diurnal cycle is found between 10 and 4 hPa. At 5 hPa the relative amplitude is 4 % (Parrish et al., 2014).

## 5.1 Chemistry of the diurnal ozone cycle

The Chapman equations describe the diurnal cycle in a static atmosphere with oxygen chemistry only. To simulate a realistic behaviour of the daily ozone variations the dynamics of the atmosphere and catalytic cycles need to be taken into account. In the description of the ozone chemistry SD-WACCM accounts for the Chapman reactions and for reactions with the following catalysts: $NO$, $NO_2$, $H$, $OH$, $HO_2$, $Cl$, $Br$.

The presence of a robust diurnal cycle in the stratosphere during polar day when the sun is above the horizon for the whole day encourages to investigate the chemical and photochemical reaction rates contributing to the ozone production and loss. These reaction rates from SD-WACCM are shown in Fig. 14 for summer solstice and equinox 2016. The reaction rates are averaged over the same intervals as in Fig. 10 and Fig. 13. The net production rate is the sum of the net chemical production rate and the contribution from the dynamics.

The Chapman reactions are summed up to one contribution because production (R2) and losses (R3, R4) are large but almost balance each other. In general the Chapman equations dominate the shape of the diurnal variation at equinox. At summer solstice however the reaction rate from the Chapman equations is always positive. The ozone losses, necessary for a diurnal cycle, result from catalytic reactions.

In the stratosphere the main ozone losses apart from the Chapman reaction R3 and R4 are due to $NO$ and $Cl$. These reaction rates itself encounter a diurnal variation. At summer solstice the ozone production exceeds the losses around noon. However during "night" the combined ozone losses are larger than the production. At equinox the reaction rate of the Chapman equations is positive during day but exceeded by the losses through $NO$ and $Cl$. During the night the net chemical ozone production is zero. We find that in the stratosphere the net chemical production is positive if the solar zenith angle is smaller than 68°. Contributions from the dynamics are diminishing during polar day whereas they are the major contribution at equinox. The stratospheric morning maximum around equinox (Fig. 10, between 2 and 8 hPa for SD-WACCM) is due to dynamics (not shown).

The main ozone loss in the stratopause region is due to $Cl$, $H$ and $NO$. At summer solstice the Chapman reaction rate encounters a minimum in the early morning where the losses exceed the production. This results in a net loss during early



morning and to a net production in the afternoon and evening. At equinox the Chapman production rate is negative at sunrise and sunset.

In the mesosphere the ozone production is fully balanced by the losses through hydrogen during polar day. At equinox the Chapman reactions are the main contribution to the diurnal cycle.

## 6 The tertiary ozone maximum as a kind of a super diurnal cycle

The tertiary ozone maximum was first reported by Marsh et al. (2001). It is a night time maximum in ozone volume mixing ratio observed in the winter middle mesosphere at 72 km and close to the polar night terminator. Because of its location it is also known as the middle mesospheric maximum. At Ny-Ålesund the enhanced ozone VMR in the middle mesosphere lasts for the whole winter whereas ozone is depleted during summer. This makes it a kind of a super diurnal cycle with a maximum in winter and a minimum in summer (see Fig. 8 at 0.1 hPa).

The tertiary ozone layer is explained by a decrease in the production of the OH radical relative to the production of odd oxygen. The OH radical is a catalytic destructor of odd oxygen. The main source of OH in the mesosphere is the photodissociation of water vapour by solar radiation with wavelengths $\lambda < 184$ nm. Close to the polar night terminator the radiation with wavelengths $\lambda < 184$ nm is strongly attenuated because of the high optical depth of the atmosphere. Hence the production rate of OH diminishes. This decrease in the production rate of OH is not matched by a decrease in the production rate of odd oxygen. The production of atomic oxygen according to the reactions R1 and R2 is still going on because the atmosphere is still transparent for radiation in the required frequency range (184 nm $< \lambda <$ 242 nm). After sunset atomic oxygen recombines to form ozone and a night time ozone maximum builds up. Model simulations of the tertiary ozone maximum (Marsh et al., 2001; Hartogh et al., 2004; Sofieva et al., 2009) reproduced well the altitude and the latitudinal extent of the peak. The models however tended to overestimate the peak VMR. The first simulations by Marsh et al. (2001) predicted a VMR of 7 ppm which was more than twice what satellite measurements suggested with 3 ppm. Hartogh et al. (2004) came to the conclusion that a strong underestimation of the true profile by measurements because of an insufficient vertical resolution is unlikely. The results of Sofieva et al. (2009) from GOME measurements confirmed the peak VMR of 2 - 4 ppm and in simulations by WACCM the peak VMR was only overestimated by 50%.

In this section we present the GROMOS-C measurements of the tertiary ozone maximum and simulations with SD-WACCM. For the analysis of the tertiary ozone maximum the retrieval of GROMOS-C is modified. The apriori profile is an MLS climatology for September which has no tertiary ozone peak. This precaution is taken to ensure that the information about the tertiary ozone maximum comes from the measurement and not from the apriori. Additionally the GROMOS-C measurements is weighted stronger in the retrieval via a larger covariance matrix of the apriori. The covariance matrix of the apriori has diagonal elements of 0.8 ppm and a Gaussian correlation decay at neighbouring levels. To compare the SD-WACCM simulations to the measurement they are convolved with the averaging kernels of GROMOS-C.

In the measurements of GROMOS-C the tertiary ozone maximum is seen at 0.08 hPa (about 64 km) (Fig. 15, top). This is at the upper limit of the retrieval but lies still within the boundaries of a measurement response of 0.8 (indicated by the dashed



white line). The tertiary ozone maximum persists over the whole winter from beginning of October to mid March. The highest ozone VMR of up to 3 ppm are measured before and after the period of polar night. This is when the polar night terminator is close to Ny-Ålesund. The mean night time (20h - 04h) VMR at 0.8 hPa from 15th October to 1st March is 1.9 ppm.

In the unconvolved SD-WACCM data the tertiary ozone maximum is clearly visible and noticeably detached from the main ozone layer (Fig. 15, bottom) compared to the GROMOS-C measurements. The highest ozone VMR occur before and after the polar night with a mean night time VMR of 2.15 ppm at 0.04 hPa (70 km). This is about 6 km higher than measured with GROMOS-C. During polar night the mean night time VMR drops to 1.95 ppm at 0.06 hPa (68 km).

The convolved ozone time series of SD-WACCM has a tertiary ozone maximum at the same pressure level as GROMOS-C (0.8 hPa) (Fig. 15, middle). However the mean night time peak VMR from 15. October to 1. March is 1.55 ppm which is 20 % smaller than the peak VMR measured by GROMOS-C. This is in contrast to previous studies which stated an overestimation of the ozone VMR by the models.

A possible enhancement of the peak VMR in the GROMOS-C measurement due to the secondary ozone layer must be considered as the wing of the averaging kernels might extend to these altitudes, though small in extent. In order to estimate the influence of the secondary ozone layer on the third ozone maximum we took an ideal ozone profile from a SD-WACCM simulation which clearly shows the three ozone layers as seen in Fig. 4 during winter. From this profile a spectrum is generated with ARTS. The ozone profile was then retrieved using the same settings as for the GROMOS-C retrieval. When the secondary ozone layer was not present in the true profile the magnitude of the tertiary ozone peak was 15 % smaller than for the case with a full secondary ozone layer. If an ideal SD-WACCM profile with no tertiary ozone maximum is taken for generating the spectrum a tertiary ozone peak is still present in the retrieved profile which is due to the influence of the secondary ozone layer. The amplitude is however significantly lower (about 70 %) than for the case where the ideal profile shows all three ozone maximas. We conclude that the tertiary ozone maximum is a real feature in the GROMOS-C measurements.

# 7 Conclusions

The ground based microwave radiometers GROMOS-C and OZORAM provide measurements of middle atmospheric ozone from the arctic station of Ny-Ålesund. The gathered datasets extend over more than one year and have a high time resolution of up to one hour. With these datasets at hand we are able to present the first comprehensive analysis of the ozone diurnal variations in the Arctic. In this study we analysed the diurnal cycle for different insolation conditions from polar night to polar day and at different altitudes in the stratosphere and mesosphere. Further an intercomparison of the measured ozone profiles with MLS measurements and SD-WACCM simulations was performed.

The intercomparison of the ozone data from the microwave radiometers with MLS and SD-WACCM shows that instruments and model are generally consistent. The dynamically induced ozone variations are well captured by the nudged model. During summer GROMOS-C is overestimating ozone by 10 % compared to the other datasets whereas the retrieval of OZORAM oscillates during winter.





At Ny-Ålesund the insolation conditions change drastically over the course of one year affecting the photochemistry of ozone. In the mesosphere a sharp decrease in ozone of 70 % at sunrise with a subsequent recovery at sunset is seen in February. The depths of the day time ozone depletion decreases towards polar day and the depletion vanishes in May. In the stratosphere the diurnal variations start in February (at 3 hPa) with a morning maximum and an afternoon minimum in ozone VMR. The

typical afternoon maximum is seen from April on and lasts the whole period of polar day. At 3 hPa the largest amplitudes are seen in April (13 %) whereas at 10 hPa the largest amplitudes are seen at summer solstice (7 %). The diurnal cycle at the stratopause (1 hPa) shows the behaviour of both regimes the stratospheric and the mesospheric. It shows a morning minimum and an afternoon maximum in March and a depletion at noon in May and June. The pattern of the photochemically induced diurnal cycle are symmetric around summer solstice. During polar night no diurnal variations are detected at all altitudes.

Because the diurnal variations of ozone are well modelled by SD-WACCM we use the model to analyse the reaction rates of the ozone production and losses. At equinox the diurnal cycle is dominated by the production and loss rates from the Chapman equations. At summer solstice the reaction rate of the Chapman equations is always positive for the stratosphere and the mesosphere. In the stratosphere the necessary losses for a diurnal cycle result from catalytic reactions. Reactions of ozone with $NO$ and $Cl$ produce the largest losses. For a solar zenith angle larger than $68°$ the combined losses exceed the ozone

production by the Chapman reactions and a diurnal variation during polar day is possible. In the mesosphere the Chapman ozone production is balanced by losses through the reaction of ozone with hydrogen.

GROMOS-C measured the tertiary ozone maximum in the arctic middle mesosphere. The peak is found at an altitude of 64 km with a mean night time VMR of 1.9 ppm and persists during the whole winter. SD-WACCM models the peak altitude about 6 km higher in altitude and with a mean night time VMR of about 2 ppm. The convolved SD-WACCM data however match

well with the GROMOS-C measurements.

*Competing interests.* The authors declare that they have no conflict of interest.

*Acknowledgements.* Observations by GROMOS-C in Ny-Ålesund have been funded by the Swiss National Science Foundation under grant number 200020-160048. For partial funding of this work we acknowledge the BMBF Germany (Project 01LG1214A) and German Research Foundation (DFG) SFB/TR 172 Arctic Amplification: Climate Relevant Atmospheric and Surface Processes, and Feedback Mechanisms

(AC)3 in Projects B06 and E02. The authors thank the electronics workshop of the IAP and the AWIPEV team for their support during the campaign.



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





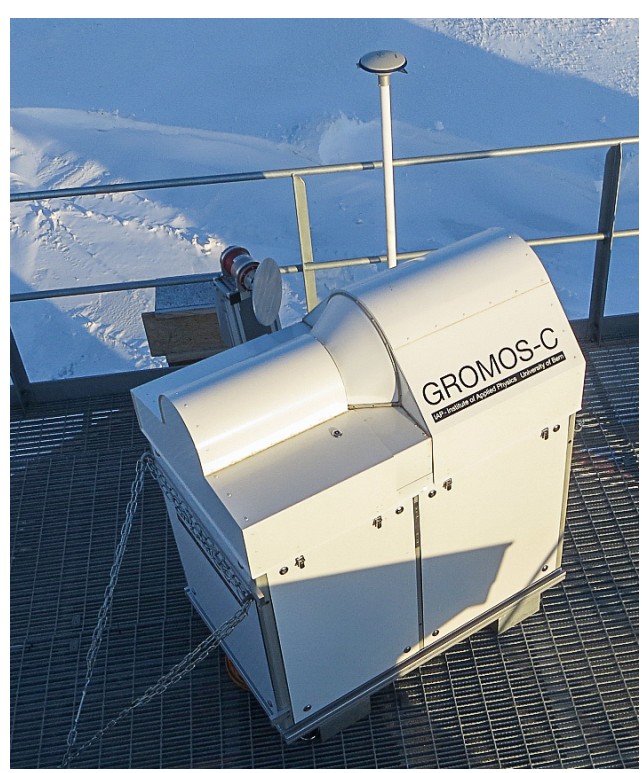

**Figure 1.** The ground based microwave radiometer GROMOS-C at the AWIPEV research base at Ny-Ålesund, Svalbard (79° N, 12° E). Observations in East and West direction are performed through a conical Teflon window with a rotating mirror inside the housing. North and Southward observations are performed via an external rotating mirror.



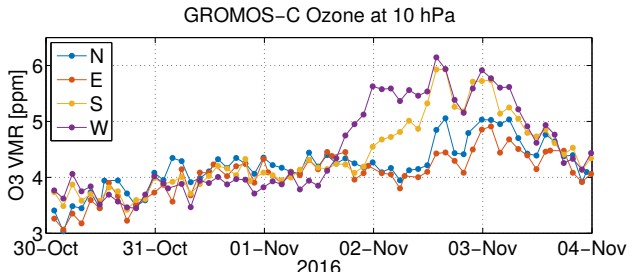

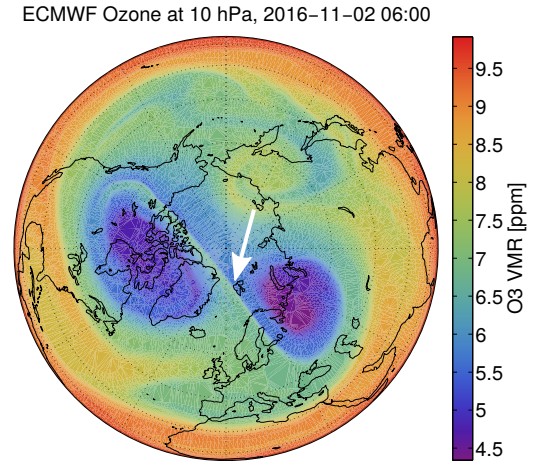

**Figure 2.** GROMOS-C ozone measurements in the four cardinal directions (N-E-S-W) at 10 hPa in November 2016 (top) and ECMWF data at 10 hPa (bottom). The arrow points to the location of Ny-Ålesund.



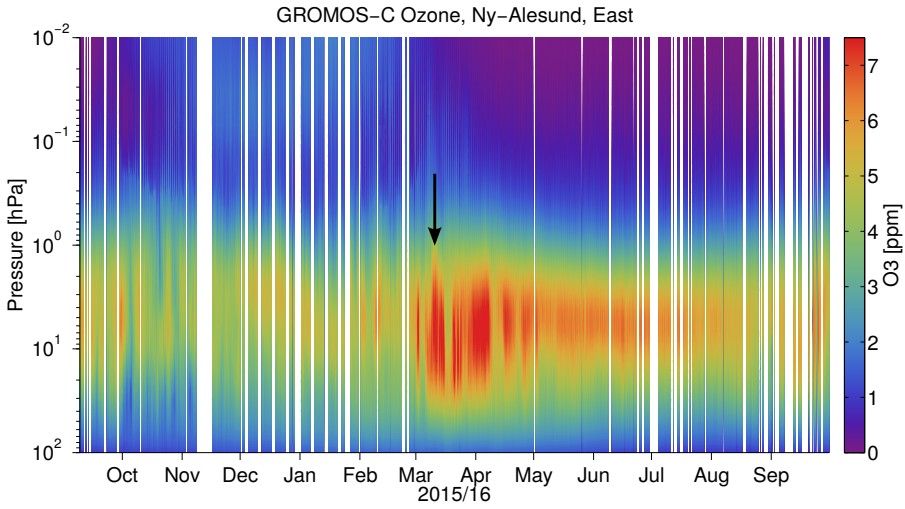

**Figure 3.** Time series of ozone volume mixing ratio measured by GROMOS-C in eastward direction at Ny-Ålesund. The arrow indicates the sudden stratospheric final warming of March 2016.

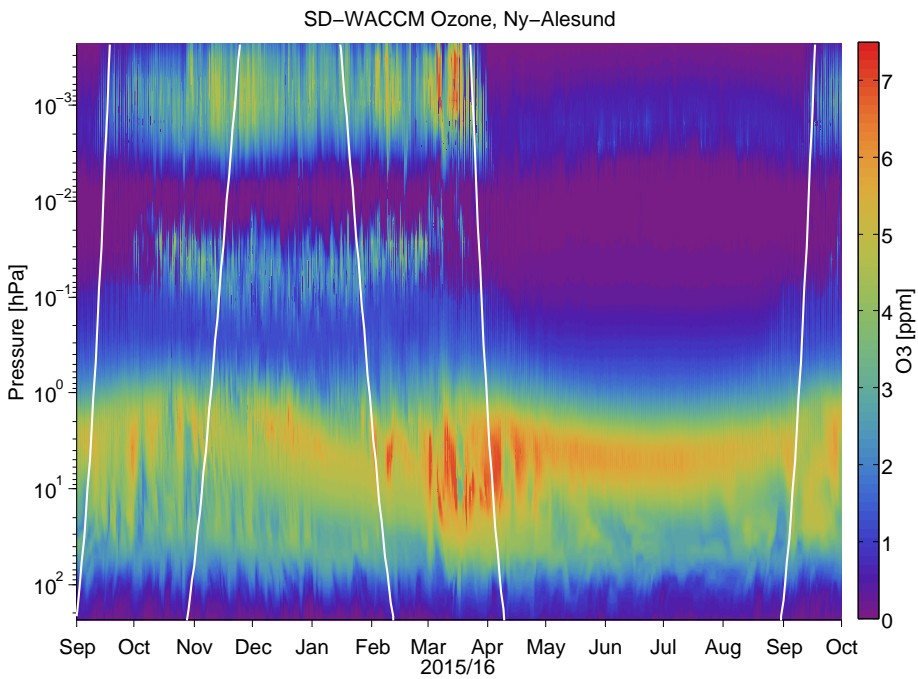

**Figure 4.** Time series of ozone volume mixing ratio modelled by SD-WACCM above Ny-Ålesund. The white lines indicate the beginning and ending of the polar day and the polar night respectively. The main ozone layer (50–1 hPa), the secondary ozone layer (0.005–0.0005 hPa) and the tertiary ozone layer (0.1–0.02 hPa) are clearly visible. During the sudden stratospheric warming in March 2016 a strong enhancement of ozone in the secondary ozone layer occurs which coincides with a decrease in the tertiary ozone layer.



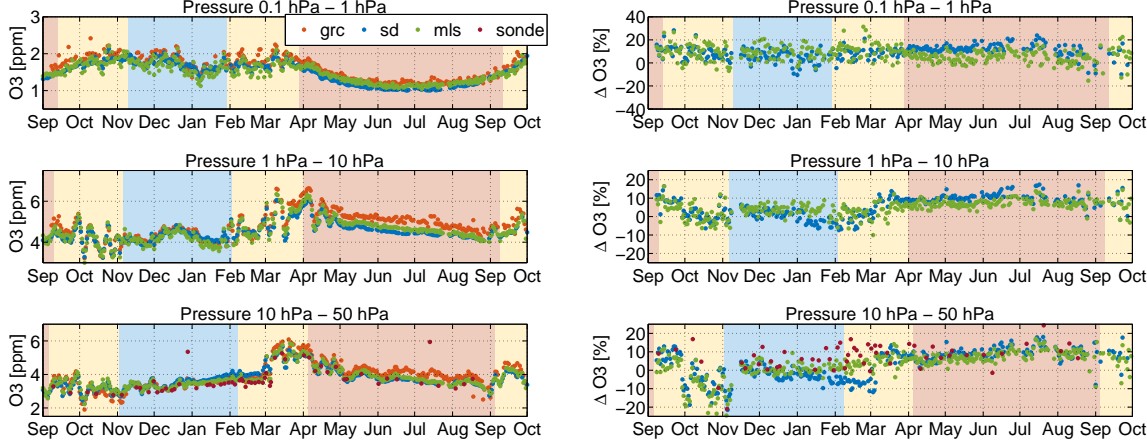

**Figure 5.** Daily mean of GROMOS-C and MLS measurements and SD-WACCM simulations averaged over three pressure intervals (left). In the lowest pressure interval additionally individual measurements from the balloon borne ozone sonde are displayed. All profiles are convolved with the averaging kernels of GROMOS-C. Relative difference between GROMOS-C measurements and the remaining datasets (right). The period of polar night is indicated by the blue background and the period of polar day by the red background. The yellow background indicates the period between polar day and night with a daily sunrise and sunset.

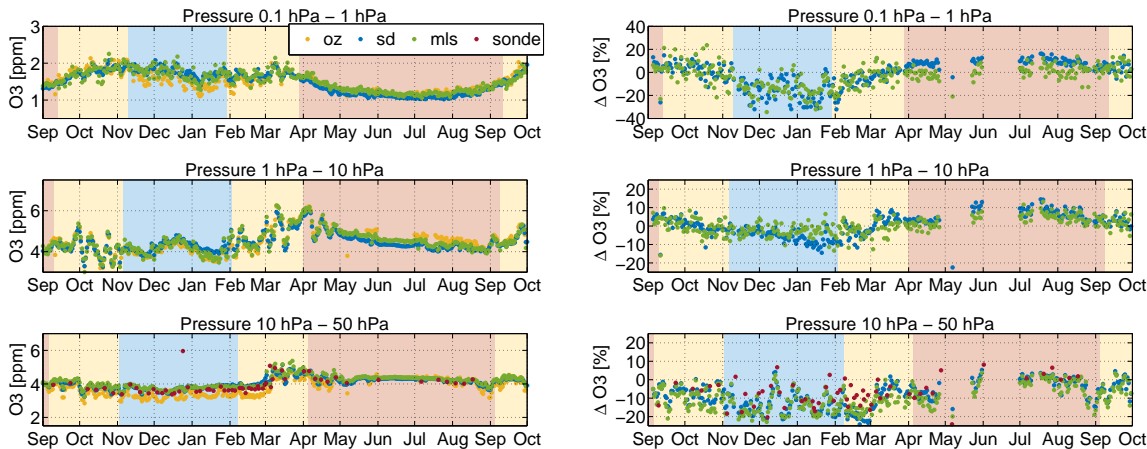

**Figure 6.** Same as in Figure 3 but for OZORAM as reference instrument.





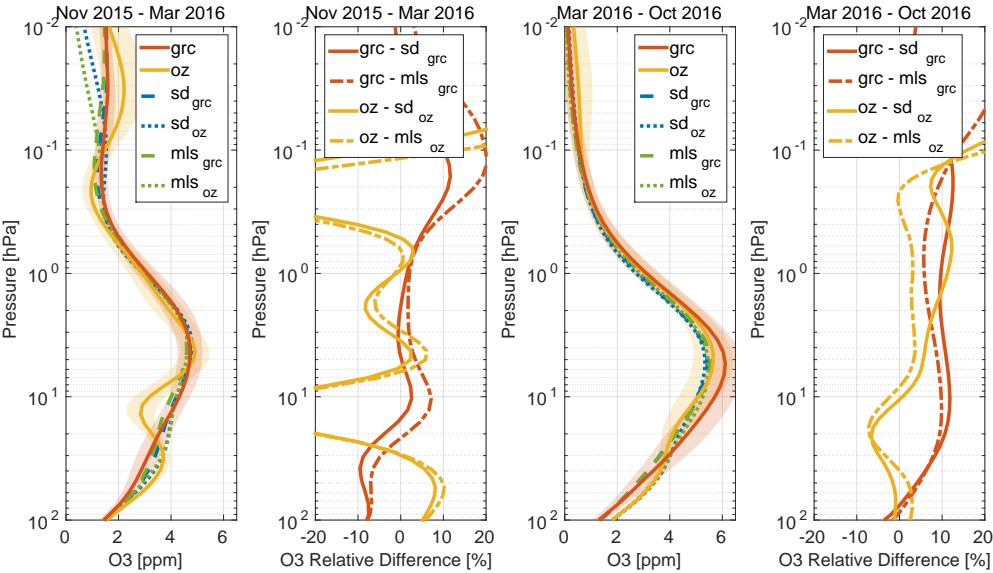

**Figure 7.** Ozone profiles of GROMOS-C, OZORAM, MLS and SD-WACCM at Ny-Ålesund averaged over a period in winter and a period in summer. The coloured bands indicate the standard deviation of GROMOS-C and OZORAM measurements. The MLS and SD-WACCM profiles are convolved and are labelled as eg. $sd_{grc}$ for a SD-WACCM profile which is convolved with GROMOS-C averaging kernels. The relative difference of the radiometer measurements to MLS and SD-WACCM are also shown.





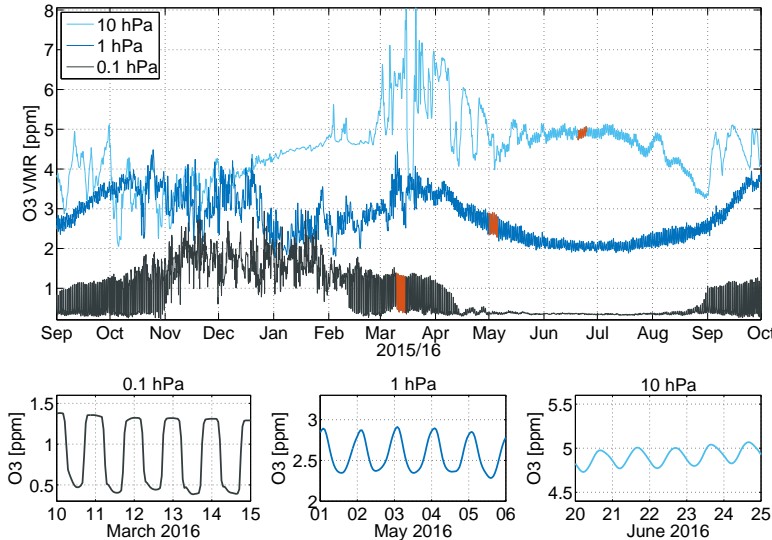

**Figure 8.** Ozone volume mixing ratio simulated by SD-WACCM at Ny-Ålesund over the course of one year and at 3 pressure levels (top). The red intervals indicate the periods for the zoomed view of the diurnal variations (bottom). Over 1 year the altitude of a fixed pressure level changes. In winter the pressure levels correspond to 28, 44 and 60 km altitude and in summer to 32, 50 and 67 km altitude





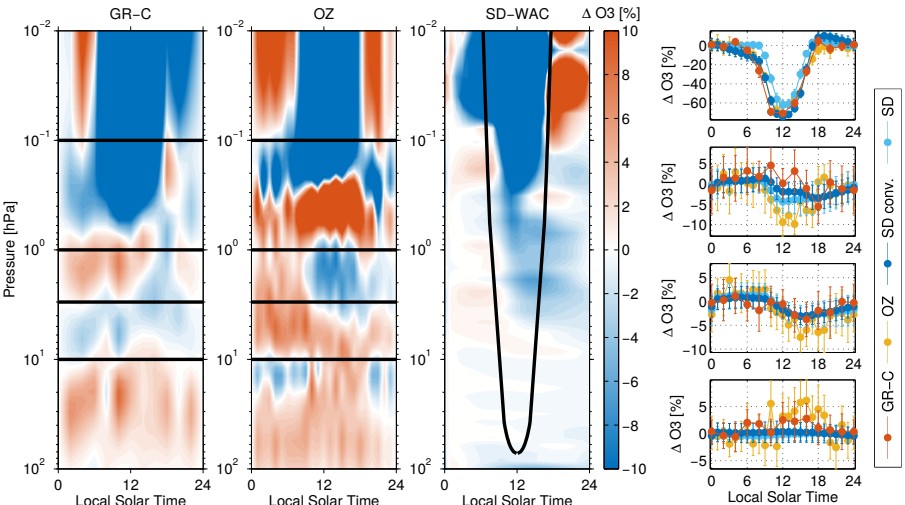

**Figure 9.** The Diurnal cycle of ozone shown as the relative difference to the midnight value from GROMOS-C measurements (1st panel), OZORAM measurements (2nd panel) and SD-WACCM data (3rd panel). The diurnal cycle is averaged over 10 days starting at 10. February 2016. The diurnal cycle is additionally shown for the convolved SD-WACCM data for 0.1 hPa, 1 hPa, 3 hPa and 10 hPa. The pressure levels are indicated by black lines in the left two panels. The black line in the 3rd panel shows the average sunrise and sunset time.

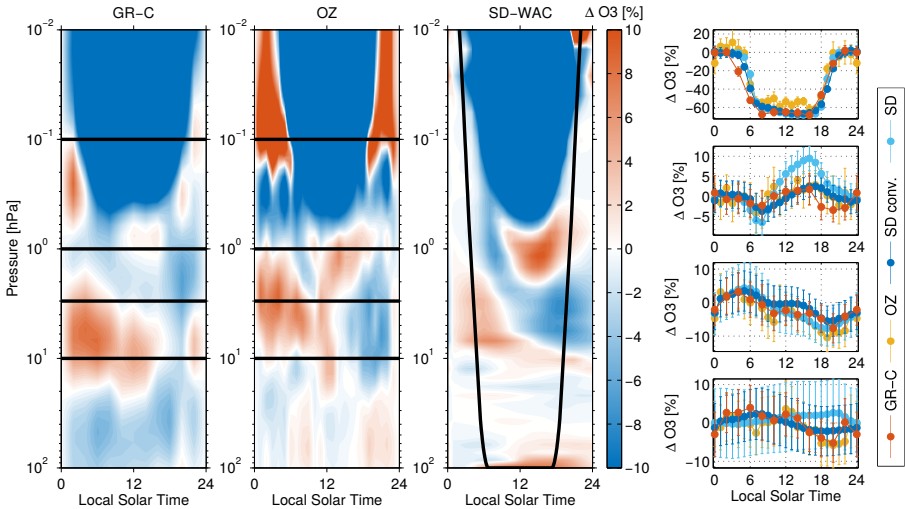

**Figure 10.** Same as in Fig. 9 but averaged over 10 days starting at 10. March 2016.



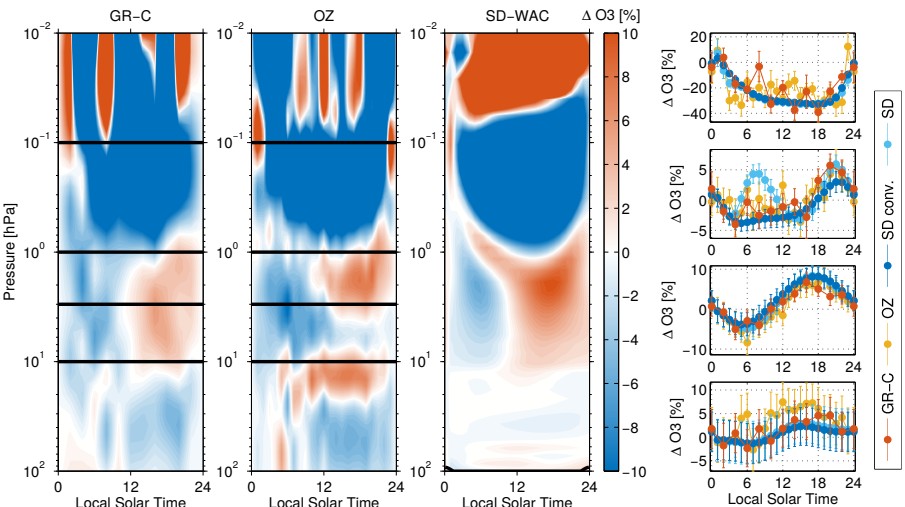

**Figure 11.** Same as in Fig. 9 but averaged over 10 days starting at 10. April 2016. Note that polar day is now at all altitudes. Remarkable is the strong diurnal cycle in the mesosphere. At the pressure level of 1 hPa two maxima exist. The morning maximum is however very narrow in altitude

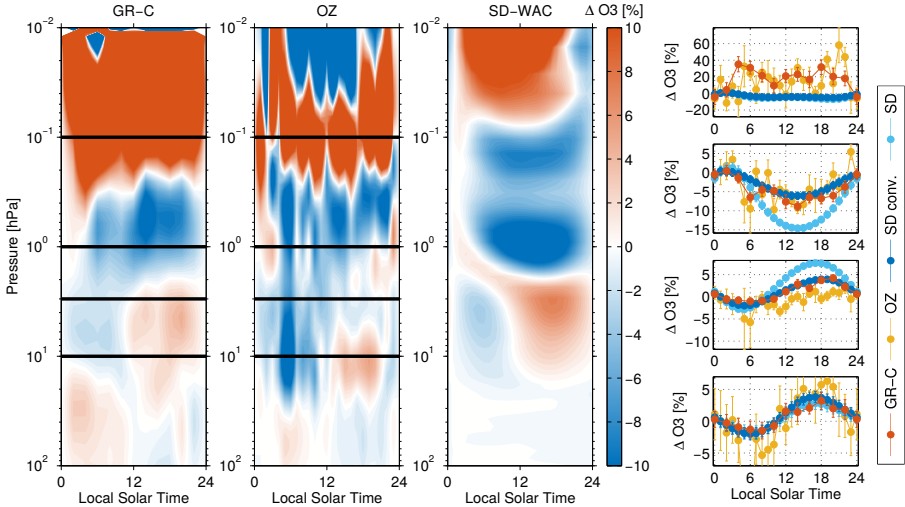

**Figure 12.** Same as in Fig. 9 but averaged over 30 days starting at 01. May 2016. The variability in the OZORAM dataset is not of atmospheric origin but is an artefact of a small dataset during May.





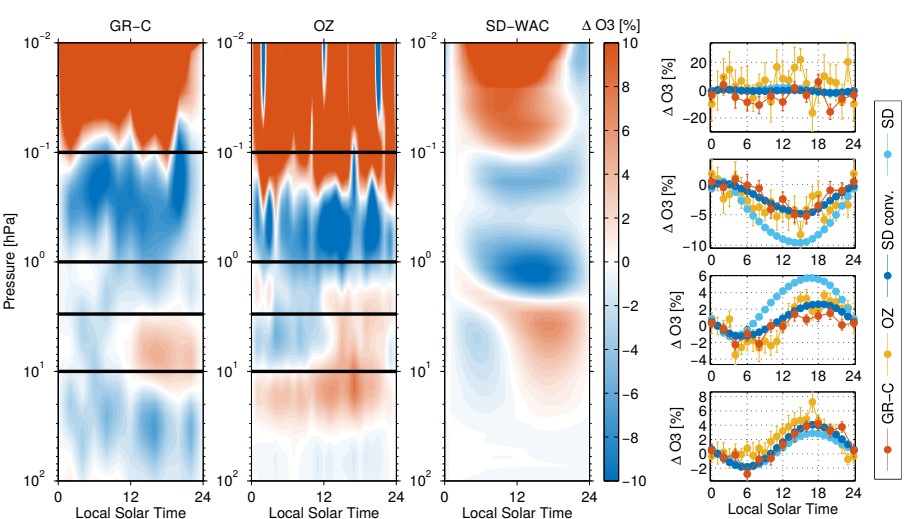

**Figure 13.** Same as in Fig. 9 but averaged over 42 days starting at 01. June 2016.





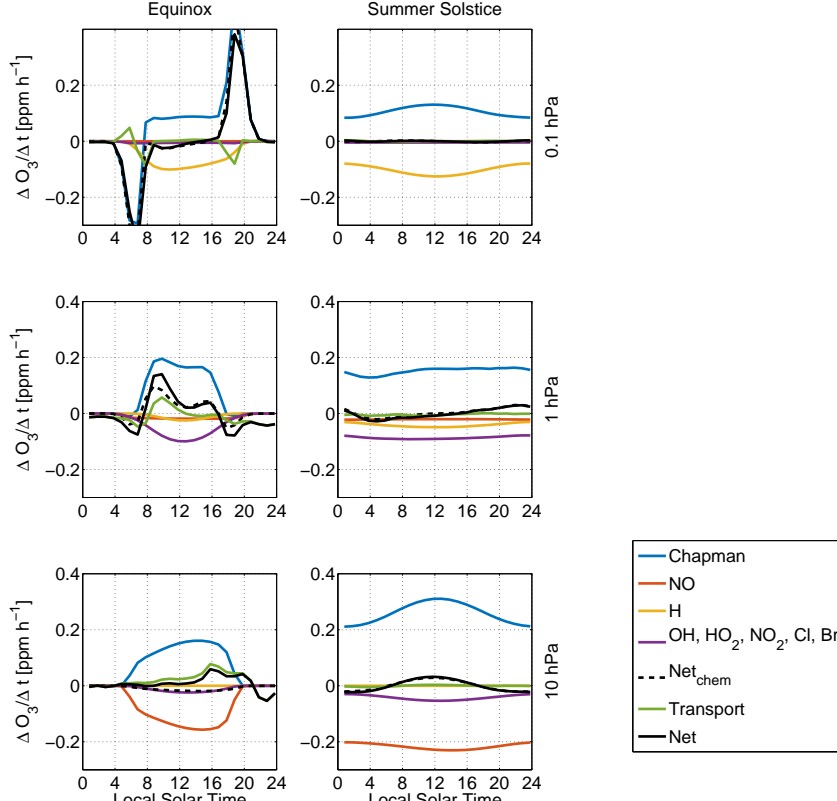

**Figure 14.** Reaction rates of ozone production and loss from SD-WACCM at Ny-Ålesund. The reaction rates are averaged over 10 days around equinox and 42 day around summer solstice of the year 2016 and are shown for 3 pressure levels. The dashed black line shows the net chemical production rate and the black solid line shows the net production rate including transport.



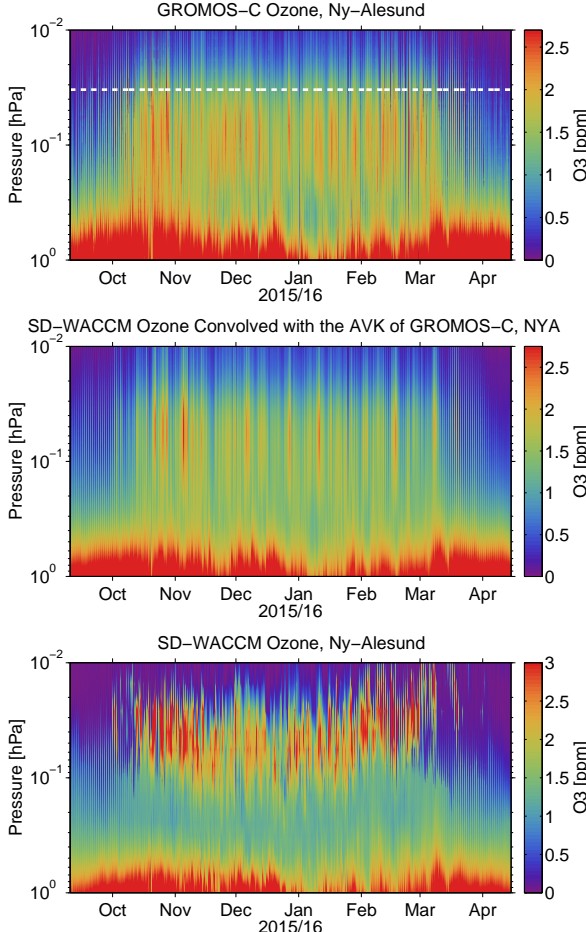

**Figure 15.** The tertiary ozone maximum at Ny-Ålesund measured by the ground based microwave radiometer GROMOS-C (top). The dashed white line indicates the measurement response of 0.8. Simulation of the tertiary ozone maximum by SD-WACCM convolved with the averaging kernels of GROMOS-C (middle) and unconvolved (bottom).





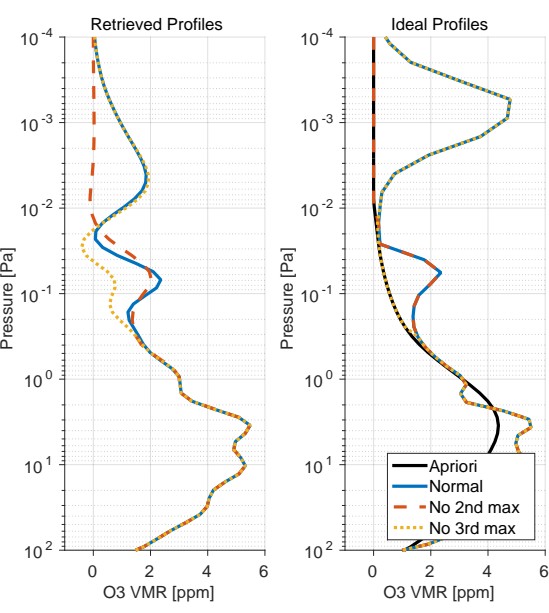

**Figure 16.** Ozone profiles (left) retrieved from spectra generated with ARTS using different ideal profiles (right). The ideal profile is from SD-WACCM and shows the three ozone maxima. It was modified to have no 2nd or no 3rd maximum.