# Peer review of "Diurnal variation in middle atmospheric ozone by ground based microwave radiometry at Ny-Ålesund over one year"

_Atmospheric Chemistry and Physics, 2017_

## Referee Comment (RC1) · B.-M. Sinnhuber (Referee) · 11 Jan 2018

The paper by Schranz and co-workers presents stratospheric and mesospheric ozone observations in the Arctic with a particular focus on the diurnal variation. Ground-based ozone observations from two different microwave radiometers are compared with simulations from the SD-WACCM model. In addition to the analysis of the diurnal cycle, intercomparisons of the two instruments between each other and with independent satellite (MLS) and ozone sonde observations are presented. I found this paper very interesting to read; it is generally well written, clearly structured and the methods seem to be sound and robust. I recommend publication in Atmos. Chem. Phys. after

consideration of the following, mostly minor, comments.

Specific comments and technical corrections:

P1L13: "task" is a funny word in this context

P1L15: not clear what the purpose of this statement is: there are also variations on shorter time scales than diurnal and longer time scales than inter-annual

P2L2: No, the other way round: photolysis becomes more important than recombination

P2L7: this sentence is a bit flawed: the magnitude of the diurnal cycle does not depend on the solar zenith angle. Do you mean variation of solar zenith angle?

P3L1 (and throughout the document): "arctic" -> "Arctic"

P3L15: "NDACC instrument" -> "OZORAM which is part of the Network for the Detection of Atmospheric Composition Change (NDACC)"

P3L20: "dynamical events" is jargon: try to be more specific

P3L25: Not sure if a historic review of Ny-Alesund is justified here. However, I do think it is relevant to provide information about the history of ground-based microwave and ozone sonde observations at Ny-Alesund.

P3L33: "very good opacity" alone not meaningful. Better give threshold opacity needed or signal-to-noise (or similar) as a function of opacity. More importantly, this is instrument specific. Would be good to briefly discuss how that affects the two microwave instruments at 110 and 142 GHz differently.

P4L26-31 and Fig.2: The discussion of different ozone observations in different viewing directions is very interesting but seems to be slightly out of place within the instrument description. I would encourage the authors (but this is not essential to this paper) to expand the discussion a bit on this point, e.g. by providing information on the difference

between ozone in different directions as a function of time.

P5L21: do you mean 1.2° latitude and 6° longitude ??

P6L16: "strength of 10%" not immediately clear: Either provide more information or just state that nudging is done up to 50km and then linearly decreasing in strength with no nudging above 60km.

P6L22: again, opacity threshold may be useful

P6L24: whether or not ozone decreases across the vortex edge depends on altitude!

P7L32: Can you give more information how the tropospheric correction is done for GOMOS-C: Even if the retrieval starts at the tropopause, tropospheric opacity has to be taken into account somehow.

P8L1: What does scaling of a standard O2 profile mean?? What is scaled? I don't think O2 is scaled.

P8L11: this statement is likely true only for this particular year. In other winters strong variations in mid-winter may be possible.

P8L21-23: why is the annual change in geopotential height relevant at this point?

P10L28: You may want to compare this to Sinnhuber et al., J. Atmos. Chem., 34, 281-290, 1999, their Fig. 7, for stratospheric ozone change as a function of solar zenith angle.

P11L9: what is a "super diurnal cycle"?

P11L23: I believe it has to be GOMOS measurements, not GOME measurements!

P12L12-21: I don't fully understand your arguments for possible differences between GROMOS-C and SD-WACCM: Any averaging kernel related effects are already taken into account when comparing with the convolved profiles, I believe?

---

## Referee Comment (RC2) · Anonymous Referee #2 · 17 Jan 2018

Schrantz et al. show that ground-based microwave remote sensing of the atmosphere has matured to a reliable and very useful technique to observe vertical profiles of diurnal variation of ozone in the middle atmosphere. The paper is interesting and well written and fulfil the scope for publication in ACP. I agree on the remarks given by referee 1 and I also recommend this paper for publication if the following two minor points are taken into consideration.

1; It would be interesting to see example spectra from the GROMOS-C and OZORAM instruments

2; Pressure scale is used to display the altitudes in the figures. I suggest that also the

altitudes in km are displayed to the left at least in figures 3, 4 and 15

---

## Author Comment (AC1) · 30 Jan 2018

**Response to Björn-Martin Sinnhuber (Referee 1)**

Franziska Schranz, Susana Fernandez, Niklaus Kämpfer, Mathias Palm

January 30, 2018

The authors are very grateful to Björn-Martin Sinnhuber (Referee 1) for carefully reading the manuscript and providing constructive comments which helped to improve the manuscript. This document contains the authors response to the specific comments and technical corrections.

1)

P1L13: task is a funny word in this context

The authors agree and the sentence has been changed.

P1L13: "Ozone is a molecule which plays an important role in the earth atmosphere."

2)

P1L15: not clear what the purpose of this statement is: there are also variations on shorter time scales than diurnal and longer time scales than inter-annual

The authors agree that the purpose of this statement is not clear and that there also exist variations on shorter and longer time scales. We modified the statement in a way that we highlight the variations which may be observed by microwave radiometers during a campaign of one year.

P1L15: "During a one-year measurement campaign with ground based microwave radiometers ozone variations can be measured on a diurnal to semi-annual time scale, depending on the instrument capability."

3)

P2L2: No, the other way round: photolysis becomes more important than recombination

Thank you! The change has been made.

P2L4: "With increasing altitude reaction R2 gets less efficient relative to R3 and more odd oxygen is stored as atomic oxygen during daytime."

4)

P2L7: this sentence is a bit flawed: the magnitude of the diurnal cycle does not depend on the solar zenith angle. Do you mean variation of solar zenith angle?

The magnitude of the diurnal cycle of ozone over a year depends on the daily mean solar zenith angle and the variation around this angle. The sentence has been adapted.

P2L8: "The magnitude of the diurnal cycle in the different altitude regimes over a year is determined by the season and the latitude with their corresponding daily mean value and variation of the solar zenith angle and length of the day."

5)

P3L1 (and throughout the document): "arctic → "Arctic

The authors agree and the change has been made throughout the document.

6)

P3L15: "NDACC instrument" → "OZORAM which is part of the Network for the Detection of Atmospheric Composition Change (NDACC)"

The change has been made.

P3L17: "The University of Bremen contributes with OZORAM (Ozone Radiometer for Atmospheric Measurements) which is part of the Network for the Detection of Atmospheric Composition Change (NDACC)."

7)

P3L20: "dynamical events" is jargon: try to be more specific

We added the specific events which we want to study during the campaign.

P3L22: "The scientific focus of the campaign lies on the investigation of dynamical events such as e.g. the dynamics of the polar vortex, sudden stratospheric warmings and planetary waves as well as on the link between middle atmospheric ozone and water vapour chemistry and the analysis of the temporal variability of those atmospheric constituents. The present study covers the temporal variability of ozone."

8)

P3L25: Not sure if a historic review of Ny-Ålesund is justified here. However, I do think it is relevant to provide information about the history of ground-based microwave and ozone sonde observations at Ny-Ålesund.

The authors see the necessity of providing information about the history of ground-based microwave and ozone sonde observations at Ny-Ålesund.

P3L30: "At this station the University of Bremen has been operating ground based microwave radiometers which measure atmospheric key parameters like water vapour (1998–2003), chlorine monoxide (1993–2003) and carbon monoxide (since 2017). Ozone is measured with OZORAM

since 1993 and ozone sondes are launched on a weekly basis since 1991."

9)

P3L33: "very good opacity" alone not meaningful. Better give threshold opacity needed or signal-to-noise (or similar) as a function of opacity. More importantly, this is instrument specific. Would be good to briefly discuss how that affects the two microwave instruments at 110 and 142 GHz differently.

We added a brief discussion of the tropospheric opacity for the two frequencies and stated that the measurement conditions for GROMOS-C (at 110.8 GHz) are ideal for an opacity < 0.5.

P4L4: "A low optical depth results in larger amplitudes of the line and reduces the integration time. For microwave radiometry as described in this paper the average optical depth at the GROMOS-C measurement frequency of 110.8 GHz was 0.7 in winter and increased to 1.2 in summer. An opacity value lower than 0.5 is considered to be ideal for such observations (Fernandez et al., 2016). OZORAM measures at 142.2 GHz which is at the wing of a water vapour line. The variability of the opacity is given by changes in water vapour and thus the line at 142.2 GHz is affected more by the variations in tropospheric water vapour than the line at 110.8 GHz (for details see Fernandez et al. (2015, Fig. 1))."

10)

P4L26-31 and Fig.2: The discussion of different ozone observations in different viewing directions is very interesting but seems to be slightly out of place within the instrument description. I would encourage the authors (but this is not essential to this paper) to expand the discussion a bit on this point, e.g. by providing information on the difference between ozone in different directions as a function of time.

We decided to keep this discussion within the instruments section. We highlight the link between the variability of ozone in the stratosphere and the location of the polar vortex edge and that changes in ozone VMR across the vortex edge can be seen in the GROMOS-C measurements. We would however not like to expand any further in this context.

P5L3:"In the stratosphere the variability of ozone can be affected by the location of the polar vortex because the ozone VMR changes dramatically across the vortex edge. This implies that the location of the vortex edge needs to be taken into account when short term ozone fluctuation like the diurnal cycle are investigated. The ozone change across the vortex edge was measured by GROMOS-C during a vortex split in the beginning of November 2016."

11)

P5L21: do you mean 1.2° latitude and 6° longitude ??

Yes we do. The correction has been made.

P5L30: "For the intercomparison with the instruments located at Ny-Ålesund MLS data are taken if the location of the measurement is within 1.2  latitude and 6  longitude from Ny-Ålesund."

12)

P6L16: "strength of 10%" not immediately clear: Either provide more information or just state that nudging is done up to 50 km and then linearly decreasing in strength with no nudging above 60 km.

We agree and the change has been made.

P6L23 : "The nudging is performed up to 50 km and then it linearly decreases in strength with no nudging above 60 km."

13)

P6L22: again, opacity threshold may be useful

A measurement is discarded if the opacity $> 1.6$ which corresponds to a transmission $< 0.2$.

P6L30: "In summer the data gaps are due to very high opacity values ($\tau > 1.6$) and strong precipitation which caused the retrieval process to fail."

14)

P6L24: whether or not ozone decreases across the vortex edge depends on altitude!

The sentence has been specified.

P7L1: "Ozone volume mixing ratios decrease sharply when the vortex passes over Ny-Ålesund at a given altitude in the stratosphere. This can be seen in Fig. 2."

15)

P7L32: Can you give more information how the tropospheric correction is done for GROMOS-C: Even if the retrieval starts at the tropopause, tropospheric opacity has to be taken into account somehow.

Applying a tropospheric correction in the case of GROMOS-C means that we calculate the brightness temperature which would be measured at the tropopause level ($T_b(z_{trop})$). Therefore we consider the troposphere as one layer with an effective temperature $T_{trop}$ and for that the simplified radiative transfer equation is

$$T_b(z_0) = T_b(z_{trop})e^{-\tau} + T_{trop}(1 - e^{-\tau}),  \tag{1}$$

where $T_b(z_0)$ is the brightness temperature measured at ground, $T_b(z_{trop})$ is the brightness temperature at the tropopause level and $\tau$ is the opacity of the troposphere. If we rearrange the equation we get

$$T_b(z_{trop}) = \frac{T_b(z_0) - T_{trop}(1 - e^{-\tau})}{e^{-\tau}}.  \tag{2}$$

The effective temperature of the troposphere is calculated from the ground temperature $T_{groud}$ according to Ingold et al. (1998)

$$T_{trop} = T_{ground} + \Delta T, \tag{3}$$

where for Ny-Ålesund at 110 GHz we take $\Delta T = -11$K. The tropospheric opacity is calculated from the wings of the measured spectrum using equation 2. At the wing we can neglect the ozone emission and $T_b(z_{trop})$ corresponds to the temperature of the cosmic microwave background $T_{bg}$. Therefore

$$\tau = -\log\left(\frac{T_{trop} - T_b(z_0)}{T_{trop} - T_{bg}}\right). \tag{4}$$

As we refer to Fernandez et al. (2015) we think it is not necessary to repeat how the tropospheric correction is done. If you consider it as important we will include the explanation in the paper.

16)

P8L1: What does scaling of a standard $O_2$ profile mean?? What is scaled? I dont think $O_2$ is scaled.

We agree that this is not correct. Not the $O_2$ profile is scaled, but the absorption coefficients obtained from a standard $O_2$ and $H_2O$ profile calculated by the MPM93 model (Liebe et al., 1993). The manuscript has been adapted accordingly.

P8L12: "In order to do this, the absorption is calculated from standard $H_2O$ and $O_2$ profiles using the MPM93 model (Liebe et al., 1993) and scaled along with the $O_3$ profile. For details refer to Palm et al. (2010)."

17)

P8L11: this statement is likely true only for this particular year. In other winters strong variations in mid-winter may be possible.

This is certainly true. The statement was adapted.

P8L21: "During this winter ozone encounters strong fluctuations due to the dynamics of the polar vortex."

18)

P8L21-23: why is the annual change in geopotential height relevant at this point?

We agree that this sentence can be deleted.

19)

P10L28: You may want to compare this to Sinnhuber et al., J. Atmos. Chem., 34, 281- 290, 1999, their Fig. 7, for stratospheric ozone change as a function of solar zenith angle.

Thank you for pointing us at this publication. We included a comparison to their Fig. 7.

P11L5: "Sinnhuber et al. (1999) used a photochemical box-model and found that for February and March the net chemical production rate is positive for solar zenith angles smaller than $80°$. We analysed the net chemical production rate from SD-WACCM according to Sinnhuber et al. (1999) for the fixed location of Ny-Ålesund and found the ozone production rate to be positive for solar zenith angles smaller than $65$–$75°$ depending on the season (Fig. 15)."

20)

P11L9: what is a "super diurnal cycle"?

We consider the variation from summer to winter time ozone VMR at the altitude of the tertiary ozone layer as a "super diurnal cycle" where a year is like a day. The sentence has been reformulated.

P11L20: "At Ny-Ålesund the enhanced ozone VMR in the middle mesosphere lasts for the whole winter whereas ozone is depleted during summer. This can be considered as a kind of a "super diurnal cycle" where a day lasts one year. The maximum VMR is seen in winter and the minimum in summer (see Fig. 8 at 0.1 hPa)."

21)

P11L23: I believe it has to be GOMOS measurements, not GOME measurements!

Yes this is true. Thank you!

P12L1: "The results of Sofieva et al. (2009) from GOMOS measurements confirmed the peak VMR of 2–4 ppm and in simulations by WACCM the peak VMR was only overestimated by 50%."

22)

P12L12-21: I dont fully understand your arguments for possible differences between GROMOS-C and SD-WACCM: Any averaging kernel related effects are already taken into account when comparing with the convolved profiles, I believe?

Yes, this is true. However we do not know if the simulation of the secondary ozone layer in SD-WACCM is correct. For example Tweedy et al. (2013) suggest that the ozone VMR at the secondary ozone layer is twice as large as simulated by SD-WACCM. Therefore the influence of the 2nd ozone layer to the measured 3rd ozone maximum could explain the difference between the measurement and the convolved simulation. We agree that this was not formulated clear enough and we adapted the paragraph. Further we added the case for an ideal profile with a doubled secondary ozone layer to Fig. 17.

P12L30: "In case of a true profile with a doubled secondary ozone layer the peak VMR of the tertiary maximum was enhanced by 15 %. If SD-WACCM substantially underestimates the VMR of the secondary ozone layer as it is suggested by Tweedy et al. (2013) this could explain the difference between the measurement and the convolved simulation."

**References**

Fernandez, S., Murk, A., and Kämpfer, N.: GROMOS-C, a novel ground based microwave radiometer for ozone measurement campaigns, Atmos. Meas. Tech., 8, 3001–3048, doi:10.5194/amt-8-2649-2015, 2015.

Fernandez, S., Rüfenacht, R., Kämpfer, N., Portafaix, T., Posny, F., and Payen, G.: Results from the validation campaign of the ozone radiometer GROMOS-C at the NDACC station of Réunion island, Atmos. Chem. Phys., 16, 7531–7543, doi:10.5194/acp-16-7531-2016, 2016.

Ingold, T., Peter, R., and Kämpfer, N.: Weighted mean tropospheric temperature and transmittance determination at millimeter-wave frequencies for ground-based applications, Radio Sci., 33, 905–918, 1998.

Liebe, H. J., Hufford, G. A., and Cotton, M. G.: Propagation modeling of moist air and suspended water/ice particles at frequencies below 1000 GHz, in: Proc. AGARD 52nd Spec. Meet. EM Wave Propag. Panel, pp. 3.1 – 3.10, Palma De Maiorca, Spain, 1993.

Palm, M., Hoffmann, C. G., Golchert, S. H. W., and Notholt, J.: The ground-based MW radiometer OZORAM on Spitsbergen  description and status of stratospheric and mesospheric O3-measurements, Atmos. Meas. Tech., 3, 1533–1545, doi:10.5194/amt-3-1533-2010, 2010.

Sinnhuber, B.-M., Müller, R., Langer, J., Bovensmann, H., Eyring, V., U., K., Trentmann, J., Burrows, J. P., and Künzi, K. F.: Interpretation of Mid-Stratospheric Arctic Ozone Measurements Using a Photochemical Box-Model, J. Atmos. Chem., 34, 281–290, 1999.

Sofieva, V. F., Kyrölä, E., Verronen, P. T., Seppälä, A., Tamminen, J., Marsh, D. R., Smith, A. K., Bertaux, J.-L., Hauchecorne, A., Dalaudier, F., Fussen, D., Vanhellemont, F., Fanton D'Andon, O., Barrot, G., Guirlet, M., Fehr, T., Saavedra, L., 1Earth, Kyr, E., Kyrölä, E., Verronen, P. T., Seppälä, A., Tamminen, J., Marsh, D. R., Smith, A. K., Bertaux, J.-L., Hauchecorne, A., Dalaudier, F., Fussen, D., Vanhellemont, F., Fanton D'Andon, O., Barrot, G., Guirlet, M., Fehr, T., and Saavedra, L.: Spatio-temporal observations of the tertiary ozone maximum, Atmos. Chem. Phys., pp. 4439–4445, 2009.

Tweedy, O. V., Limpasuvan, V., Orsolini, Y. J., Smith, A. K., Garcia, R. R., Kinnison, D., Randall, C. E., Kvissel, O.-K., Stordal, F., Harvey, V. L., and Chandran, A.: Nighttime secondary ozone layer during major stratospheric sudden warmings in specified-dynamics WACCM, J. Geophys. Res. Atmos., 118, 8346–8358, doi:10.1002/jgrd.50651, 2013.

---

## Author Comment (AC2) · 30 Jan 2018

**Response to Referee 2**

Franziska Schranz, Susana Fernandez, Niklaus Kämpfer, Mathias Palm

January 30, 2018

The authors are grateful to Referee 2 for carefully reading the manuscript and providing constructive comments which helped to improve the manuscript. This document contains the authors response to the comments of Referee 2.

1)

It would be interesting to see example spectra from the GROMOS-C and OZORAM instruments.

The quality of the spectra and the corresponding residuals are discussed in detail in the instrument description papers of GROMOS-C (Fernandez et al., 2015) and OZORAM (Palm et al., 2010). As we cite the two papers, which are freely available at the Copernicus Publications webpage, we do not think it is necessary to show the spectra in the scope of this publication.

2)

Pressure scale is used to display the altitudes in the figures. I suggest that also the altitudes in km are displayed to the left at least in figures 3, 4 and 15.

We agree that adding the altitude information in km would improve the figures. The figures 3 and 4 have therefore been modified. The plots in figure 15 (now Fig. 16) are already small and we prefer not to add additional y-axes.

**References**

Fernandez, S., Murk, A., and Kämpfer, N.: GROMOS-C, a novel ground based microwave radiometer for ozone measurement campaigns, Atmos. Meas. Tech., 8, 3001–3048, doi:10.5194/amt-8-2649-2015, 2015.

Palm, M., Hoffmann, C. G., Golchert, S. H. W., and Notholt, J.: The ground-based MW radiometer OZORAM on Spitsbergen  description and status of stratospheric and mesospheric O3-measurements, Atmos. Meas. Tech., 3, 1533–1545, doi:10.5194/amt-3-1533-2010, 2010.